# Recalibration of LBM Populations for Construction of Grid Refinement with No Interpolation

Arseniy Berezin [1,2,*], Anastasia Perepelkina [2], Anton Ivanov [2] and Vadim Levchenko [2]

1   Department of Theoretical Nuclear Physics, National Research Nuclear University "MEPhI", Kashirskoe Highway, 31, 115409 Moscow, Russia
2   Keldysh Institute of Applied Mathematics, Miusskaya Sq. 4, 125047 Moscow, Russia; mogmi@narod.ru (A.P.); aiv.racs@gmail.com (A.I.); lev@keldysh.ru (V.L.)
*   Correspondence: ArsenBRS@mail.ru

**Abstract:** Grid refinement is used to reduce computing costs while maintaining the precision of fluid simulation. In the lattice Boltzmann method (LBM), grid refinement often uses interpolated values. Here, we developed a method in which interpolation in space and time is not required. For this purpose, we used the moment matching condition and rescaled the nonequilibrium part of the populations, thereby developing a recalibration procedure that allows for the transfer of information between different LBM stencils in the simulation domain. Then, we built a nonuniform lattice that uses stencils with different shapes on the transition. The resulting procedure was verified by performing benchmarks with the 2D Poisselle flow and the advected vortex. It is suggested that grids with adaptive geometry can be built with the proposed method.

**Keywords:** lattice Boltzmann method; grid refinement; nonuniform grid; moment matching





## 1. Introduction

The lattice Boltzmann method (LBM) is one of the state-of-the-art methods in computational fluid dynamics. It is a curious example of the way the cellular automata application [1] has evolved to model fluids and was subsequently proven to be a rigorous numerical scheme for integrating the kinetic Boltzmann equation with the use of Gauss cubatures [2]. From the lattice gas stage of the method's evolution, it inherited the use of rectangular meshes, which are more common in LBM simulations than in other hydrodynamic schemes, such as the finite element and discontinuous Galerkin methods [3].

Compared with other methods in computational fluid dynamics (CFD), LBM stands out as a lightweight, efficient method with high locality and many options for parallelism [4,5]. The downside is the limited fluid parameter range within which LBM, in its classical formulation, works. It is known to lack Galilean invariance, to work in the range of low Mach numbers, and to be valid for isothermal flows. Even in that range, many applications have been found, such as mesoscale modeling in additive manufacturing [6]. Moreover, given the popularity of the method, many advanced variations have been developed for compressible flows, thermal flows, and additional physics. Presently, LBM is used in almost every area of CFD.

In this work, we studied adaptive meshes for LBM. In a simulation of a model with several characteristic scales, where both large-scale background flow and small-scale flow variations need to be reproduced, we can use a finer mesh for the areas where it is required and reduce the computational cost by coarsening the mesh in the area with lower gradients of the fluid properties. Adaptive meshes are also used to cover the areas with fine geometry features, such as thin channels. In the flow around the geometry with a high curvature or a flat boundary that is not parallel to the Cartesian axes, the rectangular mesh has to be either sufficiently fine to resolve it or be replaced by a curvilinear mesh such that the faces of its cells are on the fluid–solid boundaries in the model. Because the classical

LBM is constructed for rectangular meshes with a mesh step and time step equal to unity in nondimensional units, in all of these cases, there are issues with information transfer between the different mesh resolutions and different mesh geometries and with writing the LBM for nonrectangular mesh.

Over the years, many researchers have solved these issues by supplementing the LBM with some other numerical methods. When the mesh is refined in some regions, the information is transferred via interpolation in space and/or time. There are two ways of refining the lattice: node-based and cell-based [7,8]. In the node-based approaches, a node is added between each pair of cells. In the cell-based approaches, a lattice node is interpreted as a cubic cell, and each cell is subdivided into smaller cubes.

In the node-based methods [7,9–11], the order of interpolation can be increased by using more nodes in the reconstruction of the populations [11,12] or through the use of information on fluid moments to increase the accuracy of the relevant quantities only [13].

Cell-based methods [8,14] have the property of mass conservation. However, in the conventional method [8], the information transfer from coarse to fine mesh, which is often referred to as 'explode', is, essentially, an interpolation with a polynomial of the order zero: the values are just copied to the nearest cells. An extension with improved accuracy was proposed in References [7,14]. The mass-conserving method is used in software such as PARAMESH [15] and waLBerla [16].

There are other methods to modify the LBM so as to work with nonuniform meshes, such as the finite difference LBM [17–19], finite volume, and finite element methods [20–24], which can be applied to curvilinear, tetrahedral, and unstructured mesh geometries; Taylor series expansion; and least-squares methods [25].

In the areas with a high space resolution, a finer time step is also desired. Therefore, interpolation in time is also used. However, it can be avoided [26,27].

The motivation for the current study is based on the following observation: As a numerical scheme for the kinetic Boltzmann equation, the LBM uses a set of discrete velocities to compute the moments of the particle distribution function with Gauss quadrature rules. Various sets can be chosen as long as there are enough points in the quadrature to integrate the polynomials of a given order [28]. The discretization of the Boltzmann equation in space-time is performed by integration in the direction of the characteristic [29]. The integration along the characteristic is a first-order scheme; but, with a change of variables, a second-order scheme is obtained. The velocity sets are chosen to form a uniform lattice in space, so that the streaming in the direction of the characteristic moves the particle from one lattice node to another.

What if we choose an arbitrary lattice that fits the spatial and temporal scales and the geometry of our model and choose a velocity set for each node in such a way that every node is connected to its neighbors? If it is possible, then no other scheme except the basic LBM would be introduced in the model, interpolation would be avoided, and the order of accuracy would depend only on the proper LBM construction.

Methods exist in which LBM adapts to a stretched [30] or nonrectangular [31] mesh. The fact that a composition of different stencils was not previously used for adaptive grids is due to the following issue: A lattice with varied geometry requires different discrete velocity sets in some nodes, and a discrete particle distribution function corresponds to each velocity. These discrete particle distribution functions are the populations that are updated in the LBM. We come to a situation where, in general, there are nodes in which the numerical scheme is constructed differently and operates on a different number of populations, and it is not known how these population sets can be converted into one another.

A well-known method of rescaling was used in the original grid refinement method [9] and was improved in subsequent works [32]. This was motivated by the change in the value of the time step, which required a change in the collision parameter that controls the viscosity, and, in turn, the nonequilibrium part of the distribution functions. The change in the quadrature rule requires further transformation.

Our solution to this issue is motivated by the success in another area of advanced LBM variations. In Reference [33], a revolutionary method of variable rescaling was introduced for the modeling of high-Mach flows. Even with classical LBM collision operators and no other modifications, flows with Mach numbers over $10^3$ were reproduced. The work set the stage for many new developments in complex flow simulations. The scheme was extended and applied to various compressible flow simulations [34–39]. Recalibrating the populations to the flow velocity temperature for the collision step is known to improve stability [40].

Here, we propose to use this method of rescaling for the construction of nonuniform grids. Based on the moment matching condition from Reference [33], we develop a population recalibration method that allows streaming between LBM nodes in which the numerical parameters used in the construction of the LBM method (time and spatial scales and the velocity set) are different. In a general case, we identify the numerical parameters that are used to discretize the Boltzmann equations, and we find a method to convert a set of LBM populations between a pair of LBM nodes in which the LBM is constructed with different parameters.

The remainder of this paper is structured as follows: In Section 2, we remind the reader of the construction of the LBM method and what is required to recalibrate LBM variables from one set to another. In Section 3, we propose an algorithm for LBM simulation on a nonuniform grid with the use of the proposed recalibration method. In Section 4, we evaluate the resulting method for the trial problems; in Section 5, we discuss the impact of the results.

## 2. Theoretical Background

### 2.1. Lattice Boltzmann Method and Its Parameters

The kinetic Boltzmann equation is

$$\frac{\partial f}{\partial t} + \boldsymbol{\xi}\frac{\partial f}{\partial \boldsymbol{r}} = \Omega, \tag{1}$$

where $f(t, \boldsymbol{r}, \boldsymbol{\xi})$ is the single particle distribution function; and its arguments are the time $t$, the spatial coordinate $\boldsymbol{r}$, and the particle velocity $\boldsymbol{\xi}$. $\Omega$ is the collision term.

The macroscopic flow parameters—such as density $\rho(t, \boldsymbol{r})$, flow speed $\mathbf{u}(t, \boldsymbol{r})$, and temperature $T(t, \boldsymbol{r})$— are the moments of $f(t, \boldsymbol{r}, \boldsymbol{\xi})$:

$$\rho = \int\limits_{\mathbb{R}^D_{\xi}} f d^D\xi, \qquad \rho\mathbf{u} = \int\limits_{\mathbb{R}^D_{\xi}} \boldsymbol{\xi} f d^D\xi, \qquad \rho(u^2 + DT) = \int\limits_{\mathbb{R}^D_{\xi}} \xi^2 f d^D\xi, \tag{2}$$

where $D$ is the number of spatial dimensions, and the integral is taken over the entire velocity space $\mathbb{R}^D_{\xi}$.

In equilibrium, the collision term is zero, and the solution of (1) that satisfies (2) is

$$f^{\text{eq}}(\boldsymbol{\xi}) = \frac{\rho}{(\sqrt{2\pi}\xi_0)^D} e^{-(\boldsymbol{\xi}-\mathbf{u})^2/2\xi_0^2}, \tag{3}$$

and the temperature is $T = \xi_0^2$.

In the LBM, the integration in (2) is performed numerically. Taking the solution (3) into account, the integration kernel is chosen to be

$$\omega(\xi) = \frac{1}{(\sqrt{2\pi}\xi_0)^D} e^{-\xi^2/2\xi_0^2}. \tag{4}$$

Let us take a quadrature rule with the order of accuracy equal to an arbitrary number $n$,

$$\int_{\mathbb{R}^D} g(\boldsymbol{\xi}) d^D\xi = \int_{\mathbb{R}^D} \frac{g(\xi)}{\omega(\xi)} \omega(\xi) d^D\xi = \left/ \begin{array}{c} \boldsymbol{\xi}=\xi_0\mathbf{v} \\ d^D\xi=\xi_0^D d^Dv \end{array} \right/ =$$

$$\frac{1}{(2\pi)^{D/2}} \int_{\mathbb{R}^D} \frac{g(\xi_0\mathbf{v})}{\omega(\xi_0 v)} e^{-v^2/2} d^Dv \simeq \sum_i \frac{w_i g(\mathbf{c}_i)}{\omega(c_i)}, \quad (5)$$

where $i$ is the number of quadrature points. The discrete velocities $\mathbf{c}_i$ are defined as $\mathbf{c}_i \equiv \xi_0\mathbf{v}_i$, $\mathbf{v}_i$ are the quadrature points, and $w_i$ are the corresponding weights. The expression is exact when $g(\boldsymbol{\xi})$ is a polynomial of order $n$ or less.

With the use of (2) and (5), we deduce

$$\rho = \sum_i \frac{w_i f(t,\boldsymbol{r},\mathbf{c}_i)}{\omega(c_i)} = \sum_i f_i, \quad (6)$$

where the discrete populations $f_i$ are defined as

$$f_i(t,\boldsymbol{r}) \equiv \frac{w_i f(t,\boldsymbol{r},\mathbf{c}_i)}{\omega(c_i)} = (\sqrt{2\pi}\xi_0)^D w_i f(t,\boldsymbol{r},\mathbf{c}_i) e^{c_i^2/2\xi_0^2}. \quad (7)$$

Other moments are obtained in a similar manner:

$$\rho\mathbf{u} = \sum_i f_i\mathbf{c}_i, \qquad \rho(u^2 + DT) = \sum_i f_i c_i^2. \quad (8)$$

The equilibrium discrete populations are

$$f_i^{\mathrm{eq}} = \rho w_i e^{(2\mathbf{c}_i \cdot \mathbf{u} - u^2)/2\xi_0^2}. \quad (9)$$

The standard polynomial representation of $f_i^{\mathrm{eq}}$ [29] can be obtained by expanding this expression in the Taylor series:

$$f_i^{\mathrm{eq}} = \rho w_i \left( 1 - \frac{u^2}{2\xi_0^2} + \frac{\mathbf{c}_i \cdot \mathbf{u}}{\xi_0^2} + \frac{(\mathbf{c}_i \cdot \mathbf{u})^2}{2\xi_0^4} \right). \quad (10)$$

By inserting (10) and $T = \xi_0^2$ into (6) and (8), the moments are obtained exactly if the chosen quadrature rule is at least of the fourth order of accuracy.

In this derivation, $\xi_0$ is introduced as a scheme parameter in (4). It allows $\mathbf{c}_i$ to be scaled; therefore, it provides a connection between the temporal and spatial scales and controls the lattice geometry. Taking the expression of $f^{\mathrm{eq}}$ into account, we see that $\xi_0$ is also associated with the model temperature and the speed of sound. In the current work, we considered only the athermal LBM, and $\xi_0$ remains a numerical parameter in the scheme construction. For the athermal LBM, the fifth-order quadrature is enough [28].

The LBM equations are obtained by letting $\boldsymbol{\xi} = \mathbf{c}_i$ in (1) and multiplying (1) by $w_i/\omega(c_i)$ for each $i$:

$$\frac{\partial f_i}{\partial t} + \mathbf{c}_i \frac{\partial f_i}{\partial \boldsymbol{r}} = \Omega_i, \qquad \Omega_i \equiv \frac{w_i}{\omega(c_i)} \Omega|_{\boldsymbol{\xi}=\mathbf{c}_i}. \quad (11)$$

In this paper, the collision operator is taken in the Bhatnagar–Gross–Krook (BGK) form [41,42]:

$$\Omega = \frac{f^{\mathrm{eq}} - f}{\tau}, \qquad \Omega_i = \frac{f_i^{\mathrm{eq}} - f_i}{\tau}, \quad (12)$$

where $\tau$ is the collision parameter that controls the relaxation rate of populations and the fluid viscosity.

Let us take a uniform lattice and denote the mesh step by $\Delta x$ and the time step by $\Delta t$. Then, (11) is discretized into the two steps of the LBM:

- streaming:
$$\frac{\partial f_i}{\partial t} + \mathbf{c}_i \frac{\partial f_i}{\partial \mathbf{r}} = 0 \quad \Rightarrow$$

$$f_i(t + \Delta t, \mathbf{r} + \Delta \mathbf{r}) - f_i(t, \mathbf{r}) = 0, \quad \Delta \mathbf{r}_i \equiv \mathbf{c}_i \Delta t; \quad (13)$$

- collision:
$$\frac{\partial f_i}{\partial t} = \frac{f_i^{\mathrm{eq}} - f_i}{\tau} \quad \Rightarrow$$

$$f_i(t + \Delta t, \mathbf{r}) - f_i(t, \mathbf{r}) = \frac{\Delta t}{\tau}\left(f_i^{\mathrm{eq}}(t, \mathbf{r}) - f_i(t, \mathbf{r})\right). \quad (14)$$

In (13), the population transition vector $\Delta \mathbf{r}_i$ should point from one node to another; therefore, every velocity in the set $\{\mathbf{c}_i\}$ should be scaled accordingly. The $\xi_0$ parameter is used to scale $\{\mathbf{c}_i\}$. For example, in the widely used D2Q9 set, in which the quadrature points are

$$\left\{\mathbf{v}_i^{\mathrm{D2Q9}}\right\} = \left\{(0,0), (0,\pm\sqrt{3}), (\pm\sqrt{3},0), (\pm\sqrt{3},\pm\sqrt{3})\right\}, \quad (15)$$

this parameter is $\xi_0 = \Delta x / (\sqrt{3}\Delta t)$.

Thus, according to (13) and (14), the LBM method consists of two steps: streaming from one node to another in the direction of $\mathbf{c}_i$ and a local collision operation.

Usually, the LBM stencil D$N$Q$M$ is understood as a set of discrete velocities $\{\mathbf{c}_i\}$ corresponding to a grid that is uniform in time and space, in which $\Delta t = \Delta x = 1$. The first number in the indicated notation $N$ is the number of dimensions of space, and the second number $M$ is the number of points of the quadrature from which the stencil is built. In this paper, for convenience of presentation, we extend the standard definition of a stencil. Let us define the LBM stencil as a complete set of scheme parameters that uniquely define LBM streaming: quadrature points $\{\mathbf{v}_i\}$ and weights, scaling coefficient $\xi_0$, and time step $\Delta t$. We add these parameters to the usual stencil notation by putting them in brackets, i.e., D$N$Q$M(\Delta t, \xi_0^2)$. The commonly used D2Q9 LBM stencil (15) with $\Delta t = \Delta x = 1$ is denoted as D2Q9$(1, 1/3)$ hereafter.

The fluid viscosity $\nu$ is derived from the Chapman–Enskog analysis [29]:

$$\nu = \xi_0^2 \left(\tau - \frac{\Delta t}{2}\right). \quad (16)$$

Finally, given a lattice and a fluid with a viscosity of $\nu$, for the correctness of the classical athermal LBM with BGK collision, we have to ensure that each vector $\Delta \mathbf{r}_i$ points exactly from one lattice node to another, that the quadrature set has enough points to integrate with the fifth order of accuracy, and that (16) is satisfied.

### 2.2. Recalibration of Populations

In an LBM with adaptive grids, lattice nodes may have varying stencils. For the information exchange between the LBM nodes that operate with different LBM stencils, we need to know how the population sets are converted into one another. To construct the method of recalibration of populations from one stencil to another, let us find what changes are required with the change in $\Delta t$, $\xi_0$ and the set of quadrature points.

Let us remark that we allow the stencil and, thus, $\mathbf{c}_i$ to vary in time and space, and the derivation of (11) requires inserting $\omega(\mathbf{c}_i)$ under the derivative [43]. There is no contradiction here. In each node, we construct the LBM as if it were on a uniform grid. In the following, we find how the information is converted between the LBM populations that are constructed with the use of different stencils.

#### 2.2.1. Recalibration with $\Delta t$

According to (13), (14), the equations of the LBM are

$$f_i(t + \Delta t, \mathbf{r} + \mathbf{c}_i \Delta t) = f_i(t, \mathbf{r}) + \frac{\Delta t}{\tau}\left(f_i^{\mathrm{eq}}(t, \mathbf{r}) - f_i(t, \mathbf{r})\right). \quad (17)$$

In other words, the populations collide in $(t, \mathbf{r})$ and travel to $(t + \Delta t, \mathbf{r} + \mathbf{c}_i \Delta t)$.

To find the relationship between the populations that travel from node $(t, \mathbf{r})$ in the direction of the velocities of different stencils, let us use the Chapman–Enskog analysis. Taking into account only the term that is linear in the Knudsen number $\epsilon$ [44], we obtain

$$f_i \simeq f_i^{\text{eq}} + \epsilon f_i^{(1)}, \qquad f_i^{(1)} = -\tau \left( \partial_t^{(1)} f_i^{\text{eq}} + \mathbf{c}_i \partial_{\mathbf{r}}^{(1)} f_i^{\text{eq}} \right), \tag{18}$$

where we use the upper index $(1)$ to denote the first-order term in the expansions of populations and operators in terms of the Knudsen number. As for the derivatives, $\partial_t^{(1)} = \epsilon^{-1} \partial_t$, $\partial_{\mathbf{r}}^{(1)} = \epsilon^{-1} \partial_{\mathbf{r}}$; $\partial_t$ and $\partial_{\mathbf{r}}$ are shortcut notations for the time derivative and the gradient, respectively. Rewriting (18), we find the expression for nonequilibrium populations to be

$$f_i \simeq f_i^{\text{eq}} - \tau \left( \partial_t f_i^{\text{eq}} + \mathbf{c}_i \partial_{\mathbf{r}} f_i^{\text{eq}} \right) = f_i^{\text{eq}} - \tau D f_i^{\text{eq}}, \tag{19}$$

where the differential operator is $D \equiv \partial_t + \mathbf{c}_i \partial_{\mathbf{r}}$.

In what follows, we denote the quantities related to the coarse grid by the subscript $c$ and those related to the fine grid by the subscript $f$. Let us consider the conversion between stencils that differ only in $\Delta t$: D2QM$(\Delta t_c, \xi_0)$ and D2QM$(\Delta t_f, \xi_0)$. By inserting (19) into (17), we have

$$f_{i,c}(t + \Delta t_c, \mathbf{r} + \mathbf{c}_i \Delta t_c) - f_i^{\text{eq}}(t, \mathbf{r}) = (\Delta t_c - \tau_c) D f_i^{\text{eq}}(t, \mathbf{r}) \tag{20}$$

and a similar relationship for the fine stencil. By equating $D f_i^{\text{eq}}(t, \mathbf{r})$ in the expression for the fine and coarse stencils, we obtain a well-known relationship [9]:

$$\frac{f_{i,c}(t + \Delta t_c, \mathbf{r} + \mathbf{c}_i \Delta t_c) - f_i^{\text{eq}}(t, \mathbf{r})}{f_{i,f}(t + \Delta t_f, \mathbf{r} + \mathbf{c}_i \Delta t_f) - f_i^{\text{eq}}(t, \mathbf{r})} = \frac{\Delta t_c - \tau_c}{\Delta t_f - \tau_f}. \tag{21}$$

The relationship between $\tau_c$ and $\tau_f$ is obtained by requiring the invariance of fluid viscosity (16):

$$\nu = \xi_0^2 \left( \tau_c - \frac{\Delta t_c}{2} \right) = \xi_0^2 \left( \tau_f - \frac{\Delta t_f}{2} \right) \qquad \Rightarrow \qquad \frac{2\tau_c - \Delta t_c}{2\tau_f - \Delta t_f} = 1. \tag{22}$$

### 2.2.2. Recalibration with Both $\Delta t$ and $\xi_0$

Let us take the two stencils D$N$QM$(\Delta t_c, \xi_{0,c}^2)$ and D$N$QM$(\Delta t_f, \xi_{0,f}^2)$, which correspond to the same quadrature $\{\mathbf{v}_i\}$ but different $\xi_0$, $\Delta t$, and $\tau$. We have

$$\{\mathbf{c}_{i,c}\} \neq \{\mathbf{c}_{i,f}\}, \qquad D_c \neq D_f, \qquad f_{i,c}^{\text{eq}} \neq f_{i,f}^{\text{eq}}, \tag{23}$$

and instead of (22), we have

$$\nu = \xi_{0,c}^2 \left( \tau_c - \frac{\Delta t_c}{2} \right) = \xi_{0,f}^2 \left( \tau_f - \frac{\Delta t_f}{2} \right) \qquad \Rightarrow \qquad \frac{2\tau_c - \Delta t_c}{2\tau_f - \Delta t_f} = \frac{\xi_{0,f}^2}{\xi_{0,c}^2}. \tag{24}$$

Taking (23) into account, (20) leads to the following expression:

$$\frac{f_{i,c}(t + \Delta t, \mathbf{r} + \mathbf{c}_{i,c} \Delta t) - f_{i,c}^{\text{eq}}(t, \mathbf{r})}{f_{i,f}(t + \Delta t, \mathbf{r} + \mathbf{c}_{i,f} \Delta t) - f_{i,f}^{\text{eq}}(t, \mathbf{r})} = \frac{(\Delta t_c - \tau_c) D_c f_{i,c}^{\text{eq}}}{(\Delta t_f - \tau_f) D_f f_{i,f}^{\text{eq}}}. \tag{25}$$

Let us find how $D f_i^{\text{eq}}$ depends on the stencil. Let us express this quantity through the derivatives of the physical quantities $\rho$ and $\mathbf{u}$. From (10) and (19), we have

$$Df_i^{\text{eq}} = w_i \epsilon \left( \partial_t^{(1)}\rho - \frac{1}{2\xi_0^2}\partial_t^{(1)}\rho u^2 + \frac{c_{i,\beta}}{\xi_0^2}\partial_t^{(1)}\rho u_\beta + \frac{c_{i,\beta}c_{i,\gamma}}{2\xi_0^4}\partial_t^{(1)}\rho u_\beta u_\gamma + \right.$$

$$\left. c_{i,\alpha}\partial_\alpha^{(1)}\rho - \frac{c_{i,\alpha}}{2\xi_0^2}\partial_\alpha^{(1)}\rho u^2 + \frac{c_{i,\alpha}c_{i,\beta}c_{i,\gamma}}{2\xi_0^4}\partial_\alpha^{(1)}\rho u_\beta u_\gamma + \frac{c_{i,\alpha}c_{i,\beta}}{\xi_0^2}\partial_\alpha^{(1)}\rho u_\beta \right), \quad (26)$$

where, for convenience, we switch to tensor notations for vectors.

The time derivatives in (26) are replaced by the space derivatives with the use of conservation laws

$$\partial_t^{(1)}\rho + \partial_\alpha^{(1)}(\rho u_\alpha) = 0, \qquad \partial_t^{(1)}(\rho u_\alpha) + \partial_\gamma^{(1)}\Pi_{\alpha\gamma}^{\text{eq}} = 0, \qquad (27)$$

where

$$\Pi_{\alpha\gamma}^{\text{eq}} = \rho u_\alpha u_\gamma + \rho\xi_0^2\delta_{\alpha\gamma}. \qquad (28)$$

The time derivative is expanded:

$$\partial_t^{(1)}\rho u_\beta u_\gamma = u_\beta\partial_t^{(1)}\rho u_\gamma + u_\gamma\partial_t^{(1)}\rho u_\beta - u_\beta u_\gamma\partial_t^{(1)}\rho. \qquad (29)$$

Finally, with the use of (27) and (28), (26) becomes

$$Df_i^{\text{eq}} = w_i \epsilon \left( -\rho\partial_\alpha^{(1)}u_\alpha + \frac{c_{i,\alpha}c_{i,\beta}}{\xi_0^2}\rho\partial_\alpha^{(1)}u_\beta - \right.$$

$$\left. \frac{c_{i,\beta}}{\xi_0^2}\partial_\gamma^{(1)}\rho u_\beta u_\gamma - \frac{c_{i,\alpha}}{2\xi_0^2}\partial_\alpha^{(1)}\rho u^2 + \frac{c_{i,\alpha}c_{i,\beta}c_{i,\gamma}}{2\xi_0^4}\partial_\alpha^{(1)}\rho u_\beta u_\gamma + \mathcal{O}(u^3) \right). \quad (30)$$

According to (18), here, we can use $\partial$ instead of $\partial^{(1)}$, and we can insert $\mathbf{c}_i = \xi_0\mathbf{v}_i$ to find how $Df_i^{\text{eq}}$ depends on $\xi_0$.

$$Df_i^{\text{eq}} = w_i\left(-\rho\partial_\alpha u_\alpha + v_{i,\alpha}v_{i,\beta}\,\rho\partial_\alpha u_\beta - \right.$$

$$\left. \cdot\frac{v_{i,\beta}}{\xi_0}\partial_\gamma\rho u_\beta u_\gamma - \frac{v_{i,\alpha}}{2\xi_0}\partial_\alpha\rho u^2 + \frac{v_{i,\alpha}v_{i,\beta}v_{i,\gamma}}{2\xi_0}\partial_\alpha\rho u_\beta u_\gamma + \mathcal{O}(u^3)\right). \quad (31)$$

The underlined terms do not depend on $\xi_0$. These terms are unchanged as long as the same quadrature rule is used. The remaining terms can be neglected in the low-compressibility limit. Therefore, the recalibration expression is similar to (21):

$$\frac{f_{i,c}(t + \Delta t, \mathbf{r} + \mathbf{c}_{i,c}\Delta t) - f_{i,c}^{\text{eq}}(t, \mathbf{r})}{f_{i,f}(t + \Delta t, \mathbf{r} + \mathbf{c}_{i,f}\Delta t) - f_{i,f}^{\text{eq}}(t, \mathbf{r})} = \frac{\Delta t_c - \tau_c}{\Delta t_f - \tau_f}, \qquad (32)$$

and the relationship between $\tau_f$ and $\tau_c$ is expressed by (24).

### 2.2.3. Recalibration with a Change in Quadrature

Let us construct the conversion method between populations in $\text{D2Q}M_1(\Delta t, \xi_0^2)$ and in $\text{D2Q}M_2(\Delta t, \xi_0^2)$.

The conversions in (21) and (32) are between the stencils of the same quadrature $\{\mathbf{v}_i\}$ but different $\tau$. Therefore, these are the conversions between LBMs with different collision operators, which are expressed by essentially different equations, and the conversions give the correct nonequilibrium part of the populations.

When the $\tau$ parameter remains the same, the moment matching condition [33–36] is used for the conversion. It is required that the change in the LBM populations does not lead to a change in the physical moments:

$$\left.\left(\sum_i c_{i,x}^p c_{i,y}^q f_i\right)\right|_{\text{D2Q}M_1(\Delta t, \xi_0^2)} = \left.\left(\sum_i c_{i,x}^p c_{i,y}^q f_i\right)\right|_{\text{D2Q}M_2(\Delta t, \xi_0^2)}. \qquad (33)$$

The number of equations in the linear system (33) depends on the order of approximation of the quadrature and its symmetries [28,29,45]. For athermal fluid physics, we require at least

$$p + q \leq 5 \qquad p, q \in \mathbb{N}_0. \tag{34}$$

Additionally, an explicit relationship can be found for $f_0$, which corresponds to the zero discrete velocity vector (7):

$$(f_0/w_0)|_{\mathrm{D2Q}M_1(\Delta t, \xi_0^2)} = (f_0/w_0)|_{\mathrm{D2Q}M_2(\Delta t, \xi_0^2)}. \tag{35}$$

A similar relationship is valid whenever stencils have a shared nonzero discrete velocity.

Relations (33) and (35) form a system of linear equations that has to be solved to find the populations of one stencil, while the populations of the other stencil are known. In general, the system can be underdetermined or inconsistent. For example, in the conversion of D2Q9 to D2Q15, we need to find 15 unknown populations from 9 computable moments. In this case, we append the systems by equating the relationships from (33) to the equilibrium moments [46].

### 2.2.4. Recalibration with the Change of Stencil

In total, for the recalibration of the populations from $\mathrm{D2Q}M_1(\Delta t_1, \xi_{0,1}^2)$ to $\mathrm{D2Q}M_2(\Delta t_2, \xi_{0,2}^2)$, the two conversions are applied.

First, the quadrature rule is fixed; and $\Delta t$, $\tau$, and $\xi_0$ are modified with the use of (24) and (32). This converts $\mathrm{D2Q}M_1(\Delta t_1, \xi_{0,1}^2)$ to $\mathrm{D2Q}M_1(\Delta t_2, \xi_{0,2}^2)$. Second, the quadrature is changed while keeping the same $\tau$ and $\xi_0$ values, and the stencil is converted to $\mathrm{D2Q}M_2(\Delta t_2, \xi_{0,2}^2)$. The two steps can be performed in any order. In any case, in the middle of the conversion, there is a virtual LBM stencil (here, $\mathrm{D2Q}M_1(\Delta t_2, \xi_{0,2}^2)$) for which neither streaming nor collision take place.

## 3. Grid Refinement Interface without Interpolation

### 3.1. Grid Geometry

The recalibration method allows us to construct a transition between grids with different time and space steps if we only construct the stencils that point to or from the lattice points exactly in the grid transition region.

Let us take a grid refinement boundary with a coarse uniform grid ($\Delta t_c = \Delta x_c = 1$) on the left side and a fine uniform grid ($\Delta t_f = \Delta x_f = 1/2$) on the right side (Figure 1):

LBM stencils can be used in the "pull" or "push" paradigms, which is relevant for conversion purposes. In the "push" method, the populations are streamed from a node according to the stencil of that node. After the streaming, the populations in any node may come from different stencils, and the set of populations is not full in some nodes. In the "pull" method, populations are converted into the stencil of the node before they are streamed into that node. The "pull" method was used in the original work [33] and in the current study. In the case of uniform grids, no stencil conversions are used, and the "pull" and "push" methods are equivalent. The difference between rescaling incoming and outgoing populations in the grid refinement method that is supplemented with interpolation is reported in Reference [32].

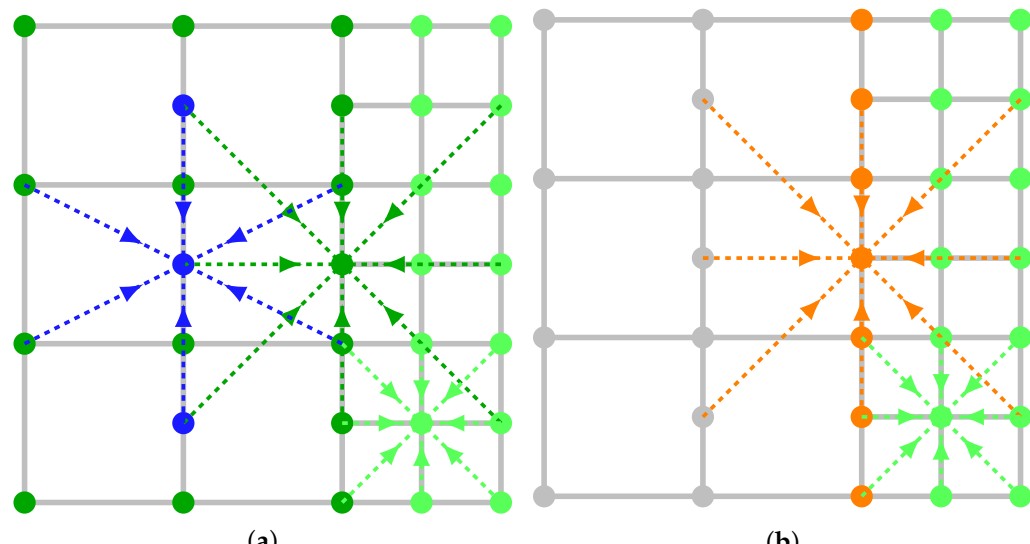

**Figure 1.** Scheme of the discrete velocity sets of the stencils near the grid refinement boundary at the integer (**a**) and half-integer (**b**) time steps. Dark-green: D2Q9(1, 1/3); light-green: D2Q9(1/2, 1/3); blue: D2Q7(1, 1/4); orange: D2Q9(1/2, 4/3); gray: no streaming at half-integer steps.

### 3.2. Stencils and Recalibration

Let us construct an LBM algorithm in which the coarse and fine grids use the classical LBM stencil, and the transition between grids does not require interpolated values; that is, all $\mathbf{c}_i$ point from one node to another. This is possible by introducing several transitional LBM stencils.

Let us use D2Q9(1, 1/3) (dark green in Figure 1) and D2Q9(1/2, 1/3) (light green) for the coarse and fine grids, respectively. This leads to the classical, unaltered D2Q9 LBM in the fine and coarse grids away from the boundary.

For the grid transition, the well-known quadrature derivation rules can be used [47]. D2Q15(1, 25/38) (see Appendix A for the derivation) is a stencil of the fifth order that uses the existing nodes. It can reproduce the moments of the equilibrium distribution at least as well as D2Q9, which is used away from the grid transition. We also tried D2Q7(1, 1/4) (details in Appendix A). This stencil performs correct integration for fewer moments, but it is more local. The tests with both stencils are reported in Section 4. In Figure 1, D2Q7(1, 1/4) is depicted, but D2Q15(1, 25/38) can be used instead.

To perform the grid transition, we use nodes with the D2Q7(1, 1/4) (or D2Q15(1, 25/38)) stencil (blue in Figure 1) on the integer time steps and the D2Q9(1/2, 4/3) stencil on half-integer time steps (orange nodes in Figure 1). This is just one of the many possibilities that can be constructed for the boundary transition. Other configurations can be used depending on the requirements of the model.

When the introduced stencils are used in the "pull" paradigm, the streaming operation at a node requests populations from the neighboring nodes. At the node from which the population is requested, recalibration into the target stencil takes place. If it is a recalibration between different variations of D2Q9, (24) is used. If any other stencil is involved, the recalibration is performed in two steps according to Section 2.2.4.

$$\text{D2Q9}(\dots) \rightleftarrows \text{D2Q9}(1,\ 1/4) \rightleftarrows \text{D2Q7}(1,\ 1/4), \tag{36}$$

$$\text{D2Q9}(\dots) \rightleftarrows \text{D2Q9}(1,\ 25/38) \rightleftarrows \text{D2Q15}(1,\ 25/38), \tag{37}$$

Here, D2Q9(1, 1/4) and D2Q9(1, 25/38) are the virtual stencils that are used only as an intermediate state in the conversion of populations.

### 3.3. Full Grid Transition Algorithm

The initial state $t = t_0$ corresponds to Figure 1a. The proposed algorithm for the grid transition is as follows:

1.  Perform streaming on the coarse grid $t_0 \to t_0 + \Delta t_c$ with the use of

    (a)     The D2Q7$(1, 1/4)$ (or D2Q15$(1, 25/38)$) stencil for the blue nodes;
    (b)     The D2Q9$(1, 1/3)$ stencil for the dark-green nodes. Here, the incoming populations at the nodes that are exactly on the boundary are saved in a separate temporary buffer to be used in Step 5, because the prestreaming populations are still needed in the next step.

2.  Perform streaming on the fine grid at $t_0 \to t_0 + \Delta t_f$ (Figure 1b) with the use of

    (a)     The D2Q9$(1/2, 1/3)$ stencil for the light-green nodes;
    (b)     The D2Q9$(1/2, 4/3)$ stencil for the orange nodes.

3.  Perform collisions on the fine grid at the orange and light-green nodes (Figure 1b).
4.  Perform the second streaming at $t_0 + \Delta t_f \to t_0 + \Delta t_c$ into the light-green nodes of the fine grid (Figure 1a).
5.  Restore the values of the boundary nodes from the buffer.
6.  Perform collisions on all nodes with the respective stencils depicted in Figure 1a.

## 4. Benchmarks

The proposed algorithm was implemented in C++ with the use of the Zipped Data Structure for Adaptive Mesh Refinement (ZAMR) [48] library in the Aiwlib [49] package. The library provides convenient tools to work with binary refined grids based on the Z-order curve traversal. It allows us to perform operations on all nodes of uneven grids and to set flags on the nodes where operations differ. With it, we set unique flags for each color of the nodes in Figure 1 and implement the described algorithm.

### 4.1. Poiseuille Flow

The stationary solution of the Navier–Stokes equation

$$\frac{\partial \mathbf{u}}{\partial t} + (\mathbf{u}\boldsymbol{\nabla})\mathbf{u} = -\frac{1}{\rho}\boldsymbol{\nabla} p + \nu\Delta\mathbf{u} \tag{38}$$

in the $-H_x/2 \le x \le H_x/2$, $0 \le y \le H_y$ domain with a no-slip boundary in $x$ and a periodic boundary in $y$:

$$\mathbf{u}\left(-\frac{H_x}{2}, y\right) = \mathbf{u}\left(\frac{H_x}{2}, y\right) = \mathbf{0}, \qquad \mathbf{u}(x, y + H_y) = \mathbf{u}(x, y), \tag{39}$$

subject to the pressure gradient

$$\boldsymbol{\nabla} p = (0, -g), \tag{40}$$

is

$$u_x = 0, \qquad u_y(x) = \frac{gH_x^2}{8\rho\nu}\left(1 - \frac{4x^2}{H_x^2}\right). \tag{41}$$

The flow is modeled with the use of the bounce-back boundary conditions on the $x$ boundary [29]. The pressure gradient is simulated with an additional term $\Delta u_y = g\tau/\rho$ in the expression for the flow velocity during the computation of the equilibrium distribution [50] for the collision step. The initial state was set by computing the equilibrium populations, in which

$$\rho = 1, \qquad u_x = u_y = 0. \tag{42}$$

The flow evolves to its equilibrium state (41).

In this benchmark, the classical LBM with BGK collision and bounce-back boundary has a well-known problem. The effective channel width $H_x$ depends on the collision parameter $\tau$ [51]. We fix $\tau_f = (\sqrt{3} + 2)\Delta t_f/4$ on the fine grid. Then, the viscosity is

$\nu = \sqrt{3}\Delta t_f/12$, and the boundary is considered to be at the distance $\Delta x_f/2$ from the boundary nodes [29].

We refine the grid near the no-slip boundaries. The sizes of the grid are

$$H_x^r = 2^{3+r}\Delta x_c + \Delta x_f, \qquad H_b^r = 2^r\Delta x_c + \frac{\Delta x_f}{2} \approx \frac{H_x^r}{8}, \qquad H_y = 4\Delta x_c, \qquad (43)$$

where $r$ is the parameter that controls the grid resolution. For $r = 1$, half of the grid geometry is as shown in Figure 2.

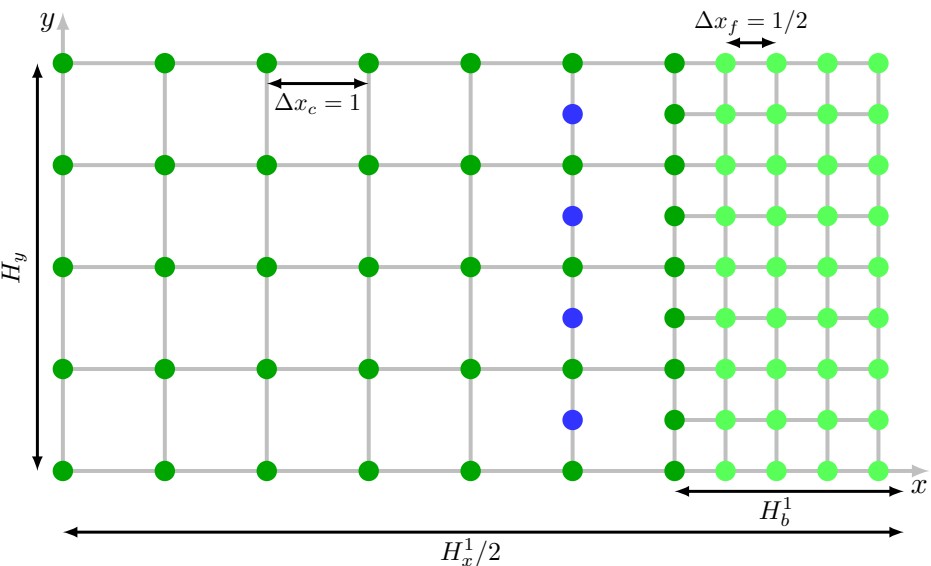

**Figure 2.** Grid geometry in the Poiseuille flow benchmark. Dark green nodes correspond to the coarse grid, light green nodes correspond to the fine grid, and blue nodes are auxiliary and use one of the stencils D2Q7(1, 1/4) or D2Q15(1, 25/38).

The fine grid takes approximately one-quarter of the domain. In our benchmarks, the tests with such grid geometry gave better results than the tests in which the grid was refined in half of the simulation region.

To evaluate the order of the approximation of the proposed scheme, the numerical error dependency on $r$ was studied. If $\Delta x$ is fixed, but $r$ is changed, the effective channel width and velocity maximum increase. Thus, the error is normalized to $(H_x^0/H_x^r)^2$. Note that in the evaluation of the $L_p$ norm in the case of nonuniform grids, the contribution of nodes is scaled to the effective area of the node $dS(\text{node})$ to ensure the correct numerical representation of the finite integral:

$$L_\infty(H_x^r) = \left(\frac{H_x^0}{H_x^r}\right)^2 \max_{\text{nodes}} |u_y^{r,\text{num}} - u_y^r|, \qquad (44)$$

$$L_n(H_x^r) = \left(\frac{H_x^0}{H_x^r}\right)^2 \sqrt[n]{\frac{\sum\limits_{\text{nodes}} |u_y^{r,\text{num}} - u_y^r|^n dS(\text{node})}{\sum\limits_{\text{nodes}} dS(\text{node})}}, \qquad n \in \{1, 2\}, \qquad (45)$$

$$dS(\text{coarse}) = \Delta x_c^2, \qquad dS(\text{fine}) = \Delta x_f^2, \qquad dS(\text{border}) = (\Delta x_c^2 + \Delta x_f^2)/2. \qquad (46)$$

The blue nodes in Figure 2 serve an auxiliary purpose in the scheme and are not taken into account in the total error.

The benchmark results are reported in Figures 3 and 4. The results are compared with those of the LBM simulation on a uniform fine grid (light green lines).

We observed that the proposed method of grid transition works correctly.

The D2Q7(1, 1/4) stencil performs better in terms of space–time discretization and produces results almost as good as those obtained with a uniform fine grid. However, the stencil does not reproduce high-order velocity moments and suffers from Galilean invariance errors. That is why its performance is better when $g$ is lower (Figure 4).

The discretization errors for the D2Q15(1, 25/38) stencil are higher and show the first order of accuracy. The source of the error has yet to be found. The dependencies of this stencil in the space–time grid are the longest of all of those of the other stencils in the constructed scheme. The low angle of the space–time characteristics may cause such behavior.

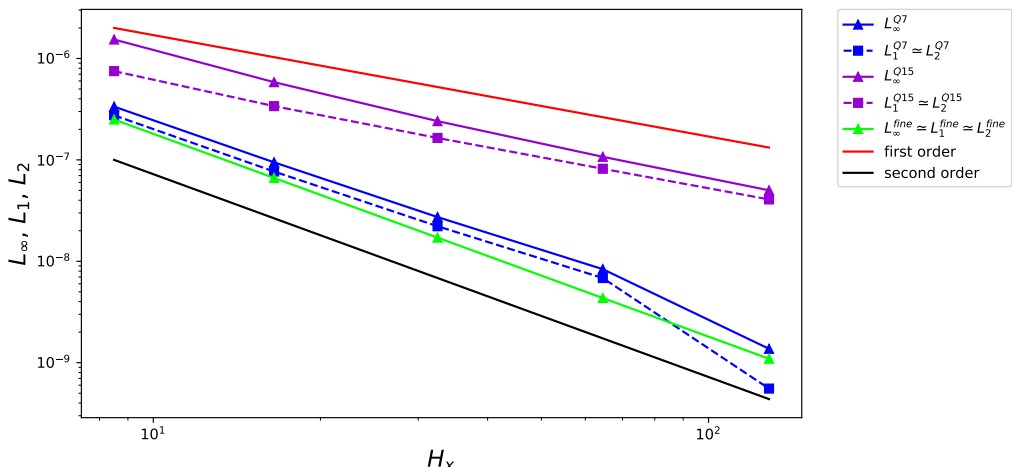

**Figure 3.** The $L_{\text{inf}}$, $L_1$ and $L_2$ errors for the Poiseuille flow with $g = 1 \times 10^{-6}$, $\nu = \sqrt{3}/24$.

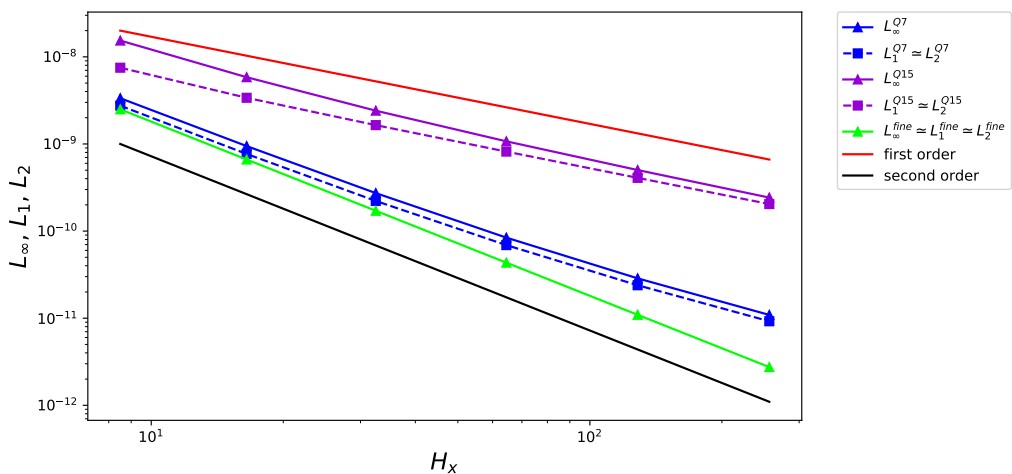

**Figure 4.** The $L_{\text{inf}}$, $L_1$ and $L_2$ errors for Poiseuille flow with $g = 1 \times 10^{-8}$ and $\nu = \sqrt{3}/24$.

Let us compare the order of accuracy of our algorithm with those of the already known schemes for constructing nonuniform grids. When the D2Q7(1, 1/4) stencil is used, we are close to the second order of accuracy for every computed norm, and this result is better than the results for both the initial-value problem (IVP) and the boundary-value problem (BVP) types of interpolation in Reference [26] and is similar to the results obtained with the cell-vertex and cell-centered hybrid-recursive regularized (HRR) algorithms with linear explosion in Reference [7]. When the D2Q15(1, 25/38) stencil is used, we obtain an order of accuracy close to unity, which is close to the results of the method with IVP interpolation in Reference [26] and the cell-centered HRR algorithm with uniform explosion in Reference [7].

Let us note that the benchmark results can be improved by the use of advanced collision and boundary conditions [29,51]. Here, we demonstrate only the impact of the proposed recalibration method; thus, the classical LBM with BGK collision is used.

In Figure 5, the difference between the numerical and theoretical solutions is plotted vs. the $x$ coordinate. Here, $r = 1$, $g = 1 \times 10^{-6}$, and $\nu = \sqrt{3}/24$. It can be seen that the grid transition introduced an error that leads to nonphysical current in the $x$-axis direction; and the error, indeed, appears on the grid boundary and is not caused by the bounce-back boundary.

In the current study, we concentrated on the verification of the method in terms of accuracy and did not optimize the code in terms of performance. However, we can say that one full LBM step, corresponding to $\Delta t = 1$, performs approximately two times faster with our code on a nonuniform grid than with the same code with classic LBM on a uniform fine grid. We are sure that with proper optimization of the code, this ratio can increase.

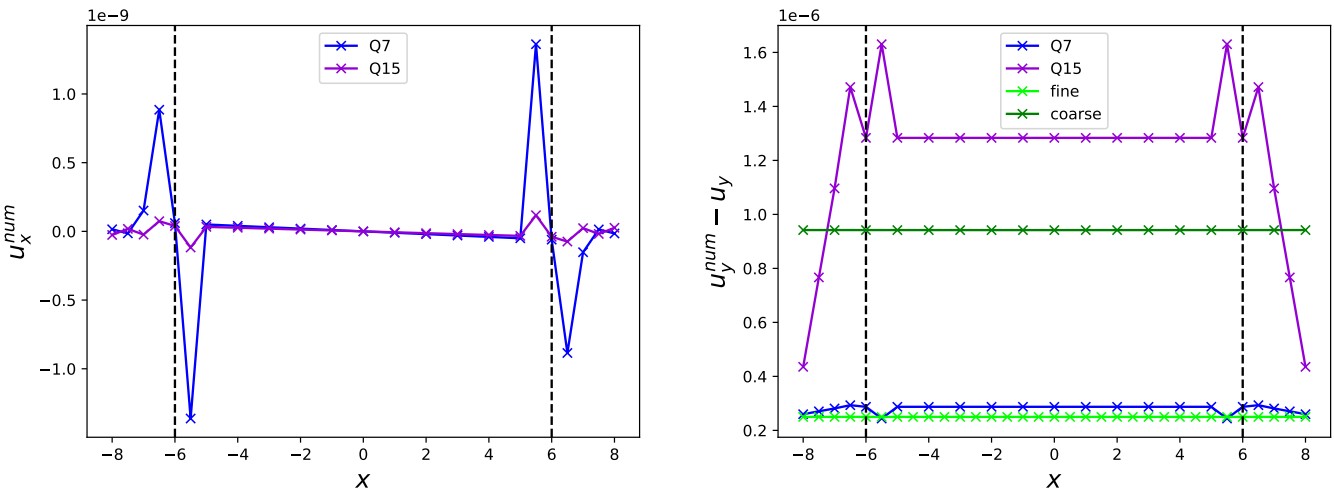

**Figure 5.** The absolute error of the Poiseuille flow solution vs. the $x$ coordinate. Here, $r = 1$, $g = 1 \times 10^{-6}$, and $\nu = \sqrt{3}/24$. Dashed lines indicate the boundaries of the transition between fine and coarse meshes.

### 4.2. Athermal Vortex

In the second benchmark, the dynamics of the advected athermal vortex [52,53] is modeled. The initial conditions are

$$u_\varphi(r)\big|_{t=0} = \frac{r\beta_0}{2\pi R} \exp\left(\frac{1 - r^2/R^2}{2}\right), \quad \rho(r)\big|_{t=0} = \exp\left(-\frac{\beta_0^2}{8\pi^2 T} \exp\left(1 - r^2/R^2\right)\right) \quad (47)$$

for the stationary vortex, and a constant $u_0$ is added to $u_x$ to model advection. Here, $\beta_0$ is the amplitude of the vortex rotation, $R$ is the vortex radius, and $T$ is the lattice temperature. According to the Navier–Stokes solution, the vortex relaxes to a uniform flow, and the relaxation rate is proportional to viscosity. In our benchmarks, we set a low viscosity and studied the preservation of the vortex shape under the influence of the grid boundaries.

The simulation domain size was $-512 \leq x \leq 512$, $0 \leq y \leq 512$, and the left half of the domain was refined. All boundaries were periodic. The vortex was initialized in the left half by setting the equilibrium populations on the fine grid with the flow parameters according to (47). The numerical parameters were $\nu = 0.01$, $\beta_0 = 0.05$, $R = 40$, $T = 1/3$, and $u_0 = 0.1$.

The D2Q15(1, 25/38) stencil was used on the grid transition. Figure 6 shows the moment of vortex interaction with the grid boundary. Its shape is preserved in the interaction, even after it passes back onto the fine grid (Figure 7). The effect of the transition on the

solution can be seen by magnifying the density range on the color map (Figure 6b). The effect is several orders of magnitude lower than the solution.

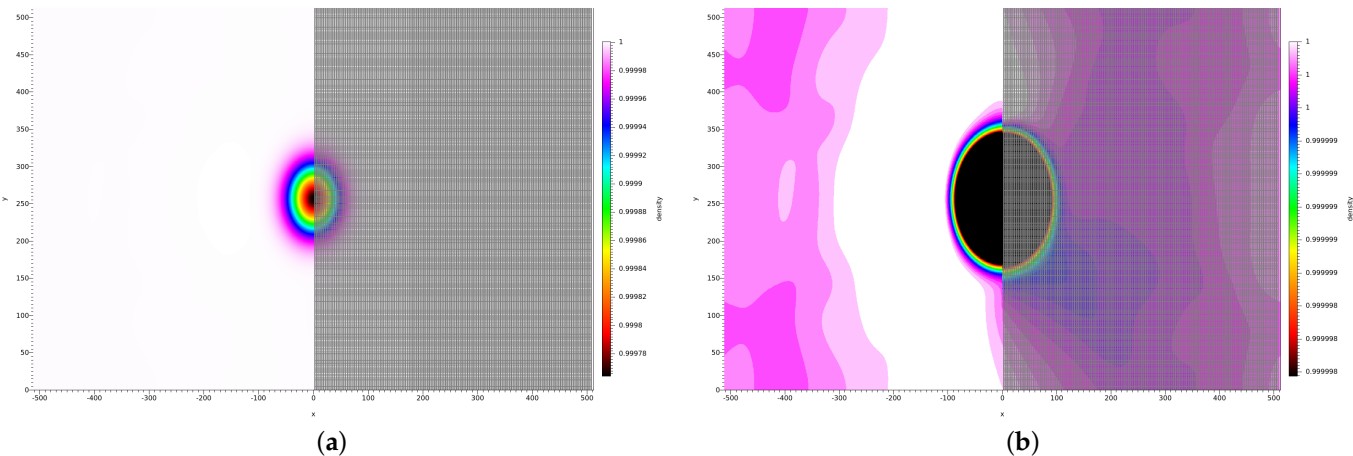

(**a**)                                               (**b**)

**Figure 6.** Vortex density distribution on the color scale at $t = 3072$. The grid on the left is refined, and empty spaces in place of the finer nodes on the coarse area are shown in gray. The left (**a**) and right (**b**) images differ in the color map range.

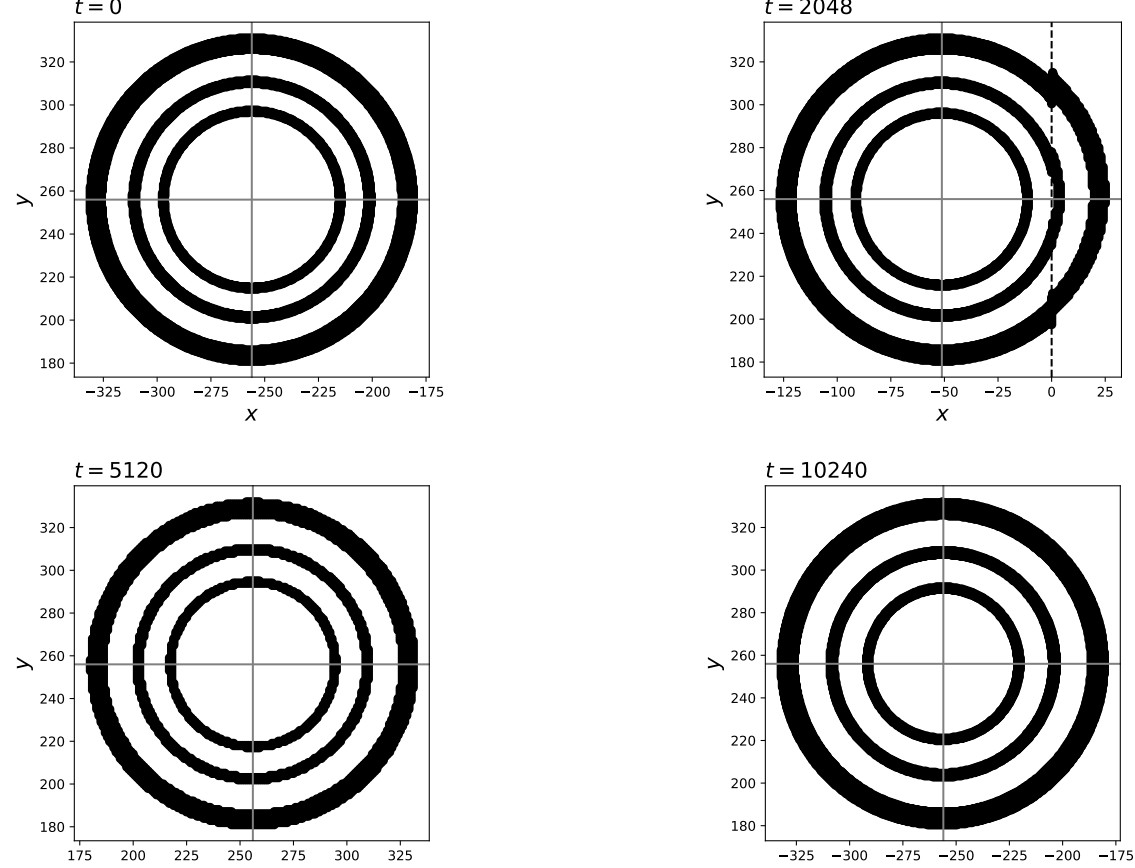

**Figure 7.** Density contours for $1 - \rho = \{75 \div 125, 375 \div 425, 875 \div 925\} \times 10^{-7}$ for 4 time instants. Solid lines indicate the center of the vortex and dashed line indicate the boundary of the transition between fine and coarse grids.

## 5. Conclusions

To conclude, we showed how the moment matching condition [11] can be used to perform conversions between LBM stencils on nonuniform grids. For an LBM with BGK collision, we developed a method of recalibrating the populations, which extends the one proposed in Reference [9] to variable lattice temperatures, and supplemented it with a method of transition to a different discrete velocity set.

We proved the recalibration to be valid on 2D benchmarks with a 1D grid transition interface. For this purpose, an algorithm that involves intermediate stencils on the transition interface was proposed.

We think that this is an important contribution because it allows building nonuniform meshes while remaining in the LBM framework, without the use of interpolation, finite differences, series expansion, or other supplementary schemes. The recalibration method is based only on the correctness of the quadrature rule in the velocity space and the Chapman–Enskog analysis. This method can be used for different configurations of mesh interfaces and even for dynamically refined or moving grids [54].

To further apply the method to more complex grid configurations, a combination of stencils for the grid interface needs to be built. After that, the proposed method can be applied to perform the transitions.

The best configuration is yet to be found. In the following studies, we may search for the source of errors that led to a worse order of approximation in the Poiseuille flow benchmark.

Furthermore, it would be interesting to formulate the recalibration procedure in the "push" framework with the rescaling of the outgoing populations [32] and moment matching of more than two population sets. By combining the "pull" and "push" types of streaming, a configuration of stencils that leads to a grid transition without mass loss may be found.

In this study, we used the basic BGK collision and the basic bounce-back boundary to focus on the impact of the recalibration method. For the other collision operators, the recalibration procedure and the adaptation of the collision parameter to the stencil may differ, but they may be built on the same principles that were discussed in the current text.

While the classical LBM is known to be limited in the range of its applicability, it is still valid in the vicinity of the equilibrium function with a fixed (usually zero) velocity, a low Mach number, and on a sufficiently detailed space–time lattice. With the use of the moment matching condition, the LBM can be locally rebuilt with different background flows [33,55,56] and different compressibility assumptions [46]. Thus, the LBM stencil can change both in time and space to adapt to the flow conditions. Our work is a demonstration of the fact that moment matching can also be used to adapt the stencil to the local geometry or to flows with a higher Reynolds number.

**Author Contributions:** Conceptualization, funding acquisition, and writing—review and editing, A.P.; methodology, validation, formal analysis, investigation, benchmarks, visualization, and writing—original draft preparation, A.B.; software, A.I.; supervision, V.L. All authors have read and agreed to the published version of the manuscript.

**Funding:** This research was funded by Russian Science Foundation grant number 18-71-10004.

**Institutional Review Board Statement:** Not applicable.

**Data Availability Statement:** Data can be provided by the authors upon request.

**Conflicts of Interest:** The authors declare no conflict of interest.

## Abbreviations

The following abbreviations are used in this manuscript:

| | |
|---|---|
| LBM | Lattice Boltzmann method |
| CFD | Computational fluid dynamics |

| | |
|---|---|
| BGK | Bhatnagar–Gross–Krook |
| ZAMR | Zipped Data Structure for Adaptive Mesh Refinement |
| IVP | Initial-value problem |
| BVP | Boundary-value problem |
| HRR | Hybrid-recursive regularized |

## Appendix A. Stencils for the Grid Transition

The stencils for the grid transitions are built according to the basic Gauss–Hermite quadrature construction rules. For the constructed quadrature rule to be no worse than the classical D2Q9 variant, we require the expression

$$\frac{1}{2\pi\xi_0^2}\int_{\mathbb{R}^2}\xi_x^p\xi_y^q e^{-(\xi_x^2+\xi_y^2)/2\xi_0^2}d^2\xi = \sum_i w_i c_{i,x}^p c_{i,y}^q, \qquad p,q\in\mathbb{N}_0, \tag{A1}$$

to be satisfied for all monomials up to the 5th order ($p+q\leq 5$).

The quadrature points $\{c_i\}$ are chosen in the grid refinement process. After that, the weights are found by solving (A1) for $w_i$ and $\xi_0$. The quantities are assumed to satisfy the standard inequalities $0 < w_i < 1$, $\xi_0 > 0$.

This amounts to a total of $(n+1)(n+2)/2 = 21$ ($n=5$) equations in general. If the quadrature point configuration is symmetrical in $x$ and $y$, the equations for all odd powers in $c_{i,x}$ or $c_{i,y}$ are trivially satisfied. A total of six equations remain to be solved. Therefore, at least five independent weights are required in addition to the free parameter $\xi_0$.

As the most local configuration with five independent shells, we choose (Figure A1)

$$\{c_i\} = \left\{(0,0),\ \left(0,\pm\frac{3}{2}\right),\ \left(\pm1,\pm\frac{3}{2}\right),\ \left(\pm1,\pm\frac{1}{2}\right),\ \left(\pm2,\pm\frac{1}{2}\right)\right\}. \tag{A2}$$

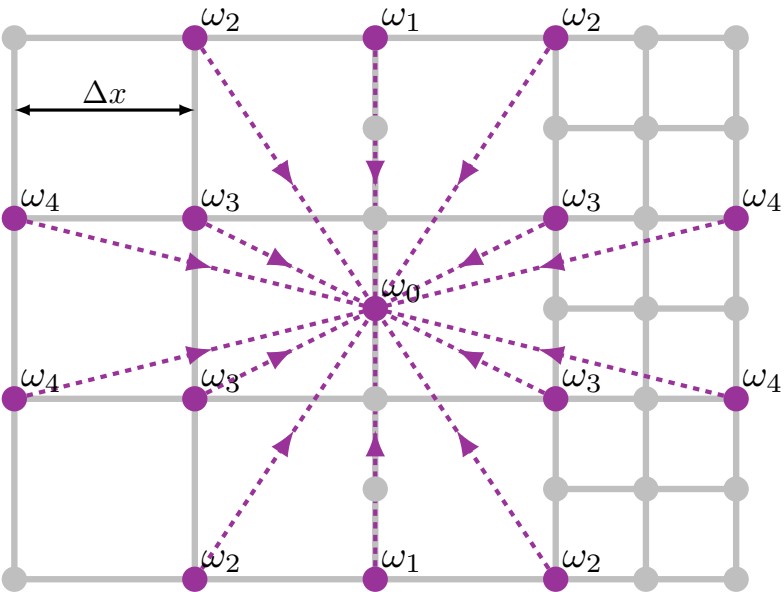

**Figure A1.** Streaming scheme for a D2Q15($\Delta t, 25\Delta x^2/(38\Delta t^2)$) stencil in the "pull" paradigm. Dashed arrows correspond to the discrete set of velocities (A2) in the case of $\Delta t = \Delta x = 1$.

The equations to be satisfied are

$$p = 0, q = 0 \qquad\qquad 1 = w_0 + 2w_1 + 4w_2 + 4w_3 + 4w_4, \tag{A3}$$

$$p = 2, q = 0 \qquad\qquad \xi_0^2 = 4w_2 + 4w_3 + 16w_4, \tag{A4}$$

$$p = 0, q = 2 \qquad\qquad \xi_0^2 = 9w_1/2 + 9w_2 + w_3 + w_4, \tag{A5}$$

$$p = 4, q = 0 \qquad\qquad 3\xi_0^4 = 4w_2 + 4w_3 + 64w_4, \tag{A6}$$

$$p = 2, q = 2 \qquad\qquad \xi_0^4 = 9w_2 + w_3 + 4w_4, \tag{A7}$$

$$p = 0, q = 4 \qquad\qquad 3\xi_0^4 = (81w_1 + 192w_2 + 2w_3 + 2w_4)/8. \tag{A8}$$

The solution of the system is

$$w_0 = \frac{1249}{3249}, \quad w_1 = \frac{6125}{103968}, \quad w_2 = \frac{775}{23104}, \quad w_3 = \frac{5375}{69312}, \quad w_4 = \frac{925}{69312}, \tag{A9}$$

and $\xi_0^2 = 25/38$.

In this work, we also use the stencil with the following points (Figure A2):

$$\{\mathbf{c}_i\} = \left\{ (0,0),\ (0,\pm 1),\ \left(\pm 1, \pm \frac{1}{2}\right) \right\}. \tag{A10}$$

For the fifth order of accuracy, the following relationships have to be satisfied:

$$p = 0, q = 0 \qquad\qquad 1 = w_0 + 4w_1 + 2w_2, \tag{A11}$$

$$p = 2, q = 0 \qquad\qquad \xi_0^2 = 4w_1, \tag{A12}$$

$$p = 0, q = 2 \qquad\qquad \xi_0^2 = w_1 + 2w_2, \tag{A13}$$

$$p = 4, q = 0 \qquad\qquad 3\xi_0^4 = 4w_1, \tag{A14}$$

$$p = 2, q = 2 \qquad\qquad \xi_0^4 = w_1, \tag{A15}$$

$$p = 0, q = 4 \qquad\qquad 3\xi_0^4 = w_1/4 + 2w_2. \tag{A16}$$

It is impossible to satisfy all of these with just four parameters, so among the fourth-order equations, we choose to satisfy only the equation for $p = q = 2$, while the equations for $p = 4$, $q = 0$ and $p = 0$, $q = 4$ are not satisfied. This is the only choice that results in a correct solution for the Poiseuille flow benchmark.

When the relationship for $p = q = 2$ is satisfied, the weights are

$$w_0 = \frac{9}{16}, \quad w_1 = \frac{1}{16}, \quad w_2 = \frac{3}{32}, \tag{A17}$$

and $\xi_0^2 = 1/4$. This is the variant used in the current work.

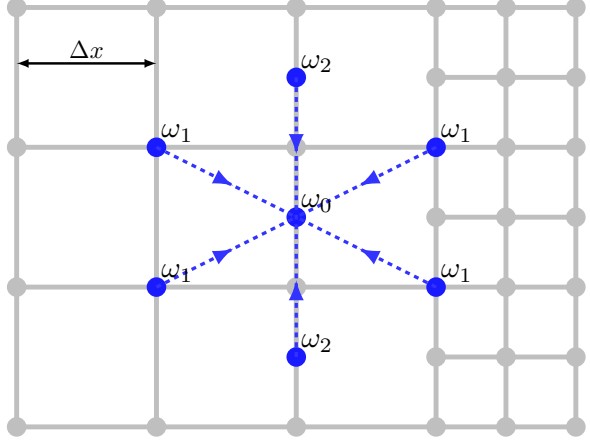

**Figure A2.** Streaming scheme for a D2Q7($\Delta t, \Delta x^2/(4\Delta t^2)$) stencil in the "pull" paradigm. Dashed arrows correspond to the discrete set of velocities (A10) in the case of $\Delta t = \Delta x = 1$.

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
