# Peer review of "Recalibration of LBM Populations for Construction of Grid Refinement with No Interpolation"

_fluids, doi:10.3390/fluids8060179_

Round 1

Reviewer 1 Report

The Authors proposed a novel approach to performing grid refinement using moment-matching conditions and rescaling the non-equilibrium part of populations. However, the main concern is to the advantage in terms of wall-clock time. Indeed, no performance numbers are reported in the manuscript. For instance, the performance should be given in terms of lattice updates per second. Also, comparing a simulation obtained with a uniform grid at the finest resolution and a mixed-resolution grid could highlight the computational trade-off due to the grid interfaces.

Minor points:

1) It would be helpful also to provide the topical stencil refinement for the 3d grid, such as d3q19.

2) How does the ZAMR library manage the address of the lattice points? Indirect addressing?

Reviewer 2 Report

There are two ways of refining the lattice: node-based and cell-based. However, the authors have developed a new grid refinement method where interpolation in space and time is not required.

In detail, moment matching condition and rescaling the non-equilibrium part of populations were used. The recalibration procedure that allows to transfer information between different LBM stencils in the simulation domain were developed.

The moment matching condition was used to perform conversions between LBM stencils on non-uniform grids.

For LBM with BGK collision, the method of recalibration of populations, which extends the to variable lattice temperatures, was developed.

The resulting procedure is verified on the 2D Poisselle flow and the advected vortex benchmark. That’s good work!

The authors also recalibration to be valid with 2D benchmarks with one-dimensional grid transition interface.

The recalibration is based on the correctness of quadrature rule in the velocity space, and the Chapman-Enskog analysis. This method can be used for different configuration of mesh interfaces, and even for dynamically refined or moving grids. I am looking forward to see those developments!

The topic is important to LBM modelers, numerical method developers, and CFD people. The topic fits the scope of the journal Fluids.

The quality of the conducted study is very high. The research background is solid. A literary review has been made quite fully. The description of models, research background are solid.

The results of this manuscript are well presented and organized. The presented results of the performed modeling method have scientific meaning.

This is an important contribution since it allows building non-uniform meshes while remaining in the LBM framework, without the use of interpolation, finite differences, series expansion, or other supplementary schemes.

Overall, the authors have done a very high level research paper with all aspects perfect.

I recommend to accept this manuscript after the minor revisions.

1.       A nomenclature of all symbols should be given.

2.       In the manuscript, a lot of words for example D2Q9 D2Q15 D2Q7 were used. It’s better to clearly explain the meaning when discussing them.

3.       Could the Aiwlib package and its Adaptive Mesh Refinement library be verified, or could the authors provide references of the verification?

4.     This work is interested for some researchers, how to apply to 3D simulation, this is a question. Besides, how to apply to some other similar fields?

5.     The format of reference is not correct.
